# Handheld Ultrasound Devices Used by Newly Certified Operators for Pneumonia in the Emergency Department—A Diagnostic Accuracy Study

**DOI:** 10.3390/diagnostics14171921

**Published:** 2024-08-30

**Authors:** Morten Jongshøj Lorentzen, Anne Heltborg Kristensen, Frida Poppius Kaldan, Mariana Bichuette Cartuliares, Mathias Amdi Hertz, Jens Juel Specht, Stefan Posth, Mats Jacob Hermansson Lindberg, Søren Helbo Skaarup, Meinhard Reinert Hansen, Camilla Stræde Spile, Michael Brun Andersen, Ole Graumann, Christian Backer Mogensen, Helene Skjøt-Arkil, Christian B. Laursen

**Affiliations:** 1Department of Emergency Medicine, University Hospital of Southern Denmark, 6200 Aabenraa, Denmarkmats.lindberg@rsyd.dk (M.J.H.L.); cbm1@rsyd.dk (C.B.M.); hsa@rsyd.dk (H.S.-A.); 2Department of Regional Health Research, University of Southern Denmark, 5000 Odense, Denmark; 3Department of Infectious Diseases, Odense University Hospital, 5000 Odense, Denmark; 4Department of Clinical Research, University of Southern Denmark, 5000 Odense, Denmark; stefan.posth@rsyd.dk; 5Department of Emergency Medicine, Odense University Hospital, 5000 Odense, Denmark; 6Department of Respiratory Diseases and Allergy, Aarhus University Hospital, 8000 Aarhus, Denmark; soerskaa@rm.dk; 7Department of Radiology, Odense University Hospital, 5000 Odense, Denmarkcamilla.straede.spile@rsyd.dk (C.S.S.); 8Department of Radiology, Copenhagen University Hospital Herlev and Gentofte, 2200 Copenhagen, Denmark; michael.brun.andersen@regionh.dk; 9Department of Clinical Medicine, Copenhagen University, 2200 København, Denmark; 10Department of Radiology, Aarhus University Hospital, 8000 Aarhus, Denmark; olegra@rm.dk; 11Department of Clinical Medicine, Aarhus University, 8000 Aarhus, Denmark; 12Department of Respiratory Medicine, Odense University Hospital, 5000 Odense, Denmark; christian.b.laursen@rsyd.dk; 13Odense Respiratory Research Unit (ODIN), Department of Clinical Research, University of Southern Denmark, 5000 Odense, Denmark

**Keywords:** diagnostic accuracy, emergency department, handheld ultrasound, lung ultrasound, novice, pneumonia

## Abstract

The diagnostic accuracy of handheld ultrasound (HHUS) devices operated by newly certified operators for pneumonia is unknown. This multicenter diagnostic accuracy study included patients prospectively suspected of pneumonia from February 2021 to February 2022 in four emergency departments. The index test was a 14-zone focused lung ultrasound (FLUS) examination, with consolidation with air bronchograms as diagnostic criteria for pneumonia. FLUS examinations were performed by newly certified operators using HHUS. The reference standard was computed tomography (CT) and expert diagnosis using all medical records. The sensitivity and specificity of FLUS and chest X-ray (CXR) were compared using McNemar’s test. Of the 324 scanned patients, 212 (65%) had pneumonia, according to the expert diagnosis. FLUS had a sensitivity of 31% (95% CI 26–36) and a specificity of 82% (95% CI 78–86) compared with the experts’ diagnosis. Compared with CT, FLUS had a sensitivity of 32% (95% CI 27–37) and specificity of 81% (95% CI 77–85). CXR had a sensitivity of 66% (95% CI 61–72) and a specificity of 76% (95% CI 71–81) compared with the experts’ diagnosis. Compared with CT, CXR had a sensitivity of 69% (95% CI 63–74) and a specificity of 68% (95% CI 62–72). Compared with the experts’ diagnosis and CT diagnosis, FLUS performed by newly certified operators using HHUS devices had a significantly lower sensitivity for pneumonia when compared to CXR (*p* < 0.001). FLUS had a significantly higher specificity than CXR using CT diagnosis as a reference standard (*p* = 0.02). HHUS exhibited low sensitivity for pneumonia when used by newly certified operators.

## 1. Introduction

Pneumonia is a major cause of morbidity and mortality for adults and children globally, with an estimated 1.3 million lives lost yearly [1]. 

Diagnosis of pneumonia can be challenging because symptoms vary considerably, and biomarkers are unspecific [2,3,4,5]. Therefore, chest X-ray (CXR) is recommended as a diagnostic tool to determine the focus of infection, even though CXRs lack diagnostic accuracy [6,7,8]. 

An alternative to CXR is bedside focused lung ultrasound (FLUS). FLUS has a diagnostic accuracy superior to CXR, costs less, and is dynamic and radiation-free [9,10,11,12]. However, it is operator-dependent, and experience is crucial for high reliability [13]. Most diagnostic accuracy studies of FLUS concerning pneumonia diagnosis were performed by experts [12,14]. However, a study of pediatric pneumonia indicated that newly certified operator’s diagnostic accuracy is significantly lower compared to experts [15]. FLUS performance by newly certified operators in adults remains uncertain.

In addition, the last few years have seen major advances in handheld ultrasound machines (HHUS), which are cheap, mobile, and quickly disinfected compared with high-end machines [16]. HHUS has promising potential, but studies into the diagnostic use of HHUS in the emergency department (ED) are limited [16]. 

Since HHUS devices are increasingly being used, and physicians with limited clinical experience often stand at the forefront in the ED, it is crucial to evaluate the diagnostic accuracy of HHUS used by newly certified FLUS operators in the ED [17].

Study objectives:

This study’s objective was to determine the diagnostic accuracy of newly certified FLUS operators’ use of HHUS for patients suspected of pneumonia in the ED, compared to CXR with expert diagnosis and CT diagnosis as the reference standard.

## 2. Materials and Methods

### 2.1. Study Design

This study was a prospective multicenter diagnostic accuracy study. Our study was part of the multifaceted Infectious Diseases in the Emergency Department (INDEED) study [18]. Patients were included from the ED of Hospital Sønderjylland’s two locations—Aabenraa and Sønderborg, at Lillebaelt Hospital and Odense University Hospital. We used convenience sampling, and research assistants included patients on weekdays between 8 a.m. and 8 p.m., from 1 February 2021 to 28 February 2022. 

Onsite researcher assistants identified potentially eligible patients using the hospitals’ logistical software (Cetrea, Getinge Cetrea A/S, Aarhus, Denmark). 

The inclusion criteria were as follows:

After an initial bedside evaluation, the treating physician considered a lower respiratory infection the most likely diagnosis.

The exclusion criteria were one or more of the following criteria being present: 

Pregnancy or age under 40 years, due to increased cancer risk from radiation.

In case of a second enrolment, the age requirement was increased to 65 years to limit accumulative radiation dose. 

Immunocompromised and patients who have been admitted for >24 h within the last 14 days of admission to accommodate the criteria for community-acquired bacterial pneumonia. 

Patients with confirmed COVID-19, as the study focused on non-COVID-19 pneumonias.

Inability or unwillingness to sign the consent form, including language barriers. 

The INDEED protocol has been published [18], and this sub-study was registered at clinicaltrials.gov (NCT 04645030) before the recruitment of patients began. This study was carried out in accordance with the declaration of Helsinki. It was approved by the Regional Committee of Health Research Ethics for Southern Denmark (record S-20200188 KH/bss), and the processing of personal data is notified to and approved by the Region of Southern Denmark and listed in the internal record (no. 20/60508) cf. Art 30 of The EU General Data Protection Regulation. Informed written consent was obtained before inclusion. 

### 2.2. Focused Lung UltraSound

The research assistants, who performed the FLUS examinations, comprised three physicians, two medical students in their final year, and one physiotherapist. None of the operators had prior FLUS experience and, therefore, underwent FLUS training and certification shortly before the inclusion by experienced trainers. The training consisted of a three-hour course with hands-on experience, followed by 25 supervised FLUS scans in the ED and culminating in a competency assessment using a validated FLUS specific objective structured assessment of ultrasound skills (OSUAS) to obtain certification, consistent with the standard training provided to Danish emergency physicians [19]. 

The operators performed the bedside FLUS using HHUS (Butterfly IQ+, Butterfly Network, Burlington, MA, USA) connected to a Smartphone (iPhone XR, Apple INC., Cupertino, CA, USA). FLUS was conducted following a 14-zone (four anterior, four lateral, and six posterior) protocol used by Laursen et al. and modified according to Volpicelli et al. [20,21]. The ultrasound preset was at the discretion of the FLUS operator. A pre-specified FLUS finding of liver-like alveolar consolidation with air bronchograms was considered diagnostic for pneumonia, as this has been shown to have high specificity for pneumonia without compromising sensitivity [12]. FLUS operators were blinded to the CT-thorax results but not to laboratory results, clinical signs, or symptoms. The treating physician was blinded to the FLUS findings. However, as a safety measure, results indicating parapneumonic effusion were shared, as these necessitated further clinical intervention. The FLUS operators were not involved in patients’ treatment, aiming to solely assess the standalone diagnostic accuracy of newly certified operators without considering the clinical context.

A convenience sample of nine patients was scanned consecutively by two FLUS operators, when schedules permitted, to assess interobserver agreement. The inter-observability scans were saved as video clips and reviewed by a lung ultrasound expert (SHS) to evaluate the agreement between FLUS operators and the FLUS expert. The FLUS expert was a pulmonologist with ten years of ultrasound experience. FLUS operators and the expert were blinded to each other’s findings. 

### 2.3. Clinical Data

CXR, blood tests, and other clinical tests were performed according to standard clinical practice at the treating physician’s discretion [22]. Clinical test results were collected from the patient’s medical records, and the research assistant collected additional medical history through patient interviews. CXR diagnoses were positive for pneumonia if a study radiologist (MRH) with 7 years’ experience concluded that pneumonia was the most likely diagnosis.

### 2.4. Reference Standard—CT-Thorax and Expert Diagnosis

A CT-thorax was performed in close proximity to the FLUS scan but not later than 24 h after admission. CT diagnosis was positive for pneumonia if the evaluating thoracic radiologist (MBA) with 14 years of experience concluded that pneumonia was the most likely diagnosis. This was based on the identification of a tree-in-bud pattern with poorly defined peribronchial nodules, as seen in bronchopneumonia, groundglass opacification or consolidation, and not in a tumor or nodular pattern.

Expert diagnosis was established by consensus between an emergency consultant and an infectious disease consultant. Medical records, blood, and microbiological test results were accessible to the experts. Results from the CT-thorax and CXR, evaluated by the radiologist on call, were also available to the experts. The experts were blinded to FLUS findings. 

Findings from the CXR and CT-thorax were evaluated separately by two separate radiologists to ensure all results from image modalities were blinded. These evaluating radiologists were blinded to the conclusions from the FLUS scans. 

### 2.5. Main Analysis

All findings and details of the patient examinations, including FLUS diagnosis and start and finish times, were recorded in Research Electronic Data Capture (REDCap) through the Open Patient Data Explorative Network (OPEN) (Odense University Hospital) in a predefined template (See Appendix A). The template did not permit an indeterminate diagnosis but did allow some FLUS zones not to be scanned. CT evaluations were recorded in a standardized form in REDCap (see Appendix A).

Descriptive data, including demographics, comorbidities, symptoms, CURB-65 score, and FLUS examination times, were reported. The mean, median, or proportion, with a corresponding 95% confidence interval (95%CI) and interquartile range, were reported as appropriate. 

The sensitivity, specificity, area under the ROC curve (AUC), positive likelihood ratio (LR+), and negative likelihood ratio (LR−) with 95%CI were calculated for FLUS and CXR individually, with CT and expert diagnosis used as the reference standards. McNemar’s test was used to compare sensitivity and specificity between FLUS and CXR. A *p*-value < 0.05 was considered significant. 

Two secondary analyses of diagnostic accuracy were calculated using CT as the reference standard to understand our findings better and allow for a more straightforward comparison with other studies. The first analysis incorporated focal b-lines as a diagnostic criterion, while the second excluded patients without a posterior FLUS scan.

Cohen’s kappa was calculated to evaluate interobserver agreement between FLUS operators and agreement between FLUS operators and the FLUS expert.

Statistical analyses were pre-specified before inclusion and performed using STATA 17, StataCorp LLC, USA. 

Patients with missing index tests were excluded from the main analysis. Patients with no available CT scan were also excluded. In addition, the data recording template did not allow for indeterminates as the study focused solely on determining the diagnostic accuracy in a vacuum, without clinical information, requiring the diagnostic criteria for pneumonia to be either present or absent. The power calculation to find pneumonia patients estimated 98% success with the reference standard and 90% success with the FLUS scanning. With a power of 80%, this study needed to include 132 patients, calculated using a one-sided McNemar test. 

## 3. Results

In total, 1101 patients with suspected pneumonia were assessed for eligibility (Figure 1). Of them, 690 patients were excluded according to general exclusion criteria, and 66 additional patients were excluded due to age-exclusion limits concerning the CT-scan. There were 94 patients excluded due to a positive COVID-19 diagnosis. Three hundred and twenty-four patients were scanned with FLUS and CT. Five patients had a CT but no FLUS due to an equipment fault during inclusion. The distribution of FLUS and expert diagnosis is shown in Figure 1.

Summary of patient characteristics for the study population is shown in Table 1. Table 2 shows FLUS and CXR diagnostic accuracy using either expert diagnosis or CT as the reference standard. The sensitivity of FLUS was significantly lower than that of CXR (*p* < 0.001). However, while there was no significant difference in specificity between the two using expert diagnosis as a reference standard (*p* = 0.4), FLUS demonstrated significantly higher specificity when CT was used as the reference standard (*p* = 0.015). 

For patients with a false-negative FLUS (n = 147) according to the expert diagnosis of pneumonia, 56 (41%) patients had right-side consolidation: 16 (11%) right upper lobe, 8 (5%) right middle lobe, and 32 (22%) right lower lobe. In addition, 41 (28%) patients had left-side consolidation: 17 (12%) left upper lobe and 24 (16%) left lower lobe. Eighty-nine (61%) patients had no pneumonic consolidation on CT, and fifty-three (36%) patients had negative CT results for pneumonia. Two CT examples of false-negative FLUS are shown in Figure 2. The expert diagnosis for false-positive FLUS is shown in Appendix A.

All FLUS zones were scanned in 235 (74%) patients. An example of a FLUS scan with consolidation and air bronchograms is shown in Figure 3. The posterior zones were not scanned for 81 (25%) patients. The median FLUS examination time was 11 (IQR 7–18) minutes. The median time between admission, FLUS, CT, and CXR is listed in Table 3 and Table 4.

Nine patients were scanned twice and used for inter- and intraobserver agreement. Cohen’s kappa score for interobserver agreement was 0 between FLUS operators, corresponding to no agreement. Cohen’s kappa was 0.06 between FLUS operators and FLUS experts, corresponding to no agreement. The tabulation of Cohen’s kappa is shown in Appendix A. 

## 4. Discussion

### 4.1. Summary of Findings

This study is the first to explore the diagnostic accuracy of FLUS for pneumonia when performed by newly certified operators using HHUS. FLUS could identify 31–32% of pneumonia patients in this study. Despite both FLUS and CXR demonstrating low sensitivity, FLUS showed significantly lower sensitivity for pneumonia than CXR when compared to either expert or CT diagnosis as a reference standard. FLUS had a higher specificity than CXR using CT diagnosis as a reference standard but comparable specificity when expert diagnosis was the reference standard. The marginal specificity advantage of FLUS over CXR was undermined by FLUS’s markedly inferior sensitivity.

### 4.2. Comparison to Other Studies

Our reported low diagnostic accuracy of HHUS by newly certified operators stands in contrast to experienced operators with high-end ultrasound machines. FLUS findings of consolidation have a reported sensitivity of 78% and a specificity of 95% for pneumonia when performed by experienced operators with high-end ultrasound [12]. It is notable that the diagnostic accuracy of CXR in our study was also lower than that reported in a meta-analysis, where both sensitivity and specificity were 75% with CT as the reference [23].

Our results show that experienced FLUS operators with high-end ultrasound machines cannot be substituted by newly certified operators using HHUS devices for diagnosing pneumonia in an ED setting. Furthermore, a large proportion of pneumonia cases might not be image-detectable at ED admission, as indicated by our findings, with 61% of false-negative FLUS scans having no pneumonic consolidation on CT and 36% CTs having no visible opacities consistent with pneumonia. However, some of these results could also stem from misclassifying infections (e.g., upper respiratory tract infections) as pneumonia among the expert diagnoses.

Our low sensitivity for detecting pneumonia can be explained by the general limitations of HHUS, the Butterfly IQ+ probe and its image quality, operator inexperience, or a combination of these factors.

Although a few studies indicate that HHUS is generally non-inferior to regular ultrasound machines, HHUS might have limited ability to detect lung consolidation [24,25,26]. Consistent with our findings, Yamanaka et al. found that HHUS had a sensitivity of 54% for consolidation on CT in admitted patients suspected of aspiration pneumonia [24]. 

Most HHUS devices investigated in the literature utilize crystals in the ultrasound probe [25]. In contrast, the Butterfly IQ+ uses a capacitive micromachined ultrasonic transducer, which few studies have compared to regular machines [16,25]. The image quality of the Butterfly IQ+ is subjectively lower than that of high-end ultrasound machines and crystal-based HHUS devices [27,28]. We found no study that compared the diagnostic accuracy for pneumonia between the butterfly IQ+ and high-end ultrasound machines. The Butterfly IQ+’s large linear contact surface might have reduced the detection of lung pathology and may be less optimal for FLUS than curved probes [29,30]. Therefore, the low diagnostic accuracy may only be related to lung ultrasound or consolidation with air bronchograms, as the Butterfly IQ+ has shown agreement with high-end ultrasound machines for other conditions, including the COVID-19 score, which includes consolidation as one of several diagnostic criteria [31,32,33,34]. However, COVID-19 typically manifests as diffuse lung involvement and pneumonia caused by other infectious agents is often localized and, therefore, easier to miss with FLUS [35,36]. 

Consolidation can be challenging to differentiate from atelectasis, pleural effusion, and interstitial syndrome, and it is a leading cause of disagreement among experts [37]. In our study, the low interobserver agreement between operators themselves and between operators and experts indicates that operator competence was still a factor. Amatya et al. and Unluer et al. focused on newly certified operators with a comparable experience level to the operators in our study and reported a sensitivity of 91–96% and a specificity of 61–84% [38,39]. However, these two studies used high-end ultrasound machines and consolidation and/or focal B-lines as diagnostic criteria for pneumonia, explaining some of the higher sensitivity [12]. When we added the criteria of focal B-lines, sensitivity increased and specificity decreased. However, our results were still significantly lower than those reported by Amatya et al. and Unluer et al. [38,39]. In addition, inexperienced operators’ diagnostic accuracy may be more affected by image quality compared to experienced operators, who can better compensate for the limitations of HHUS through factors such as the choice of preset, imaging technique, and interpretation [40,41]. 

The field of HHUS is rapidly advancing, with new devices continually introduced [25,27,42]. As technology improves, their utility will likely increase. Further testing and a framework for evaluating HHUS devices are essential for progress in identifying lung pathology.

Our study has several methodical strengths. Firstly, our FLUS training and certification were robust and easily implementable. Secondly, our two reference standards enabled the calculation of diagnostic accuracy not only for pneumonia in general but also specifically for pneumonias that are detectable through imaging upon admission.

The study’s limitations included the following:

Firstly, the 25 supervised FLUS examinations that inexperienced FLUS operators performed before inclusion may not provide sufficient experience to identify consolidations correctly [14,43,44,45]. However, all FLUS operators passed a validated structured assessment [19]. The six FLUS operators scanned 324 patients throughout the study, and therefore, each operator, on average, scanned more than 50 patients, which ought to provide an intermediate level of experience. 

Secondly, the Butterfly IQ+ is compatible with smartphones and tablets. In our study, we used a smartphone because of portability advantages. However, a tablet-size screen (larger screen than smartphone) may have assisted in improving diagnostic accuracy. 

Thirdly, reasons for missing CT scans for patients having completed a FLUS examination were not recorded. While the unavailability of timeslots in the CT scanner may have been a factor, it is plausible that deteriorating patient conditions also played a role. This has likely introduced selection bias, resulting in the sickest patients not being scanned.

Fourthly, excluding patients with COVID-19, a differential diagnosis to pneumonia, means that the diagnostic accuracy likely is lower in settings where the polymerase chain reaction for COVID-19 is not extensively available at admission [34,35].

### 4.3. Generalizability

Conducted during the COVID-19 pandemic, this study’s findings may be influenced by the unique microbiological etiology and the specific composition of the CAP population prevalent during this period. The exclusion of immunosuppressed patients, those unable to consent, and readmissions narrows the study’s generalizability. Such distinctions are not typically made at the point of admission, thus limiting the applicability of our findings to the broader pneumonia population. The inclusion criterion of suspected CAP could have skewed our sample towards more severe cases, potentially leading to an overestimated sensitivity due to more progressed consolidation. Additionally, the FLUS operators involved in our study had a proficiency level comparable to that of newly certified operators in Denmark [46]. As a result, our findings may not be applicable to settings with more advanced FLUS training, where higher diagnostic accuracy might be achieved. 

## 5. Conclusions

FLUS performed by recently certified operators in the ED using HHUS showed comparable specificity but significantly lower sensitivity for pneumonia compared to CXR. Both modalities had low diagnostic accuracy for pneumonia at the time of admission.

## Figures and Tables

**Figure 1 diagnostics-14-01921-f001:**
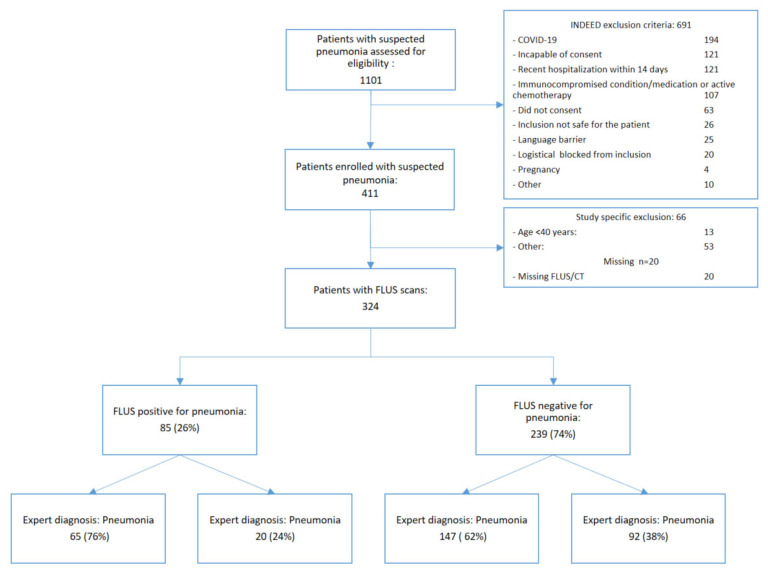
Flow chart of patients included in the study. The reference standard was expert diagnosis. FLUS = focused lung ultrasound, UTI = urinary tract infection.

**Figure 2 diagnostics-14-01921-f002:**
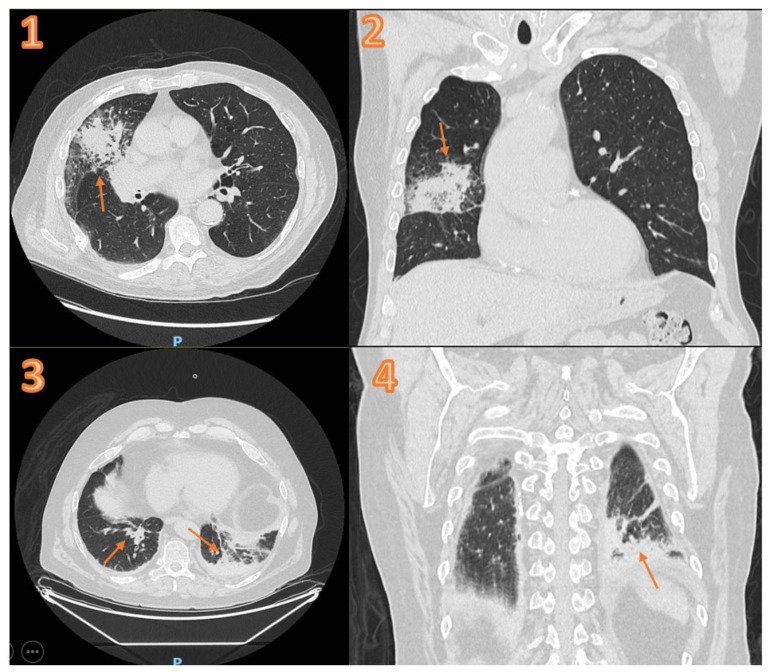
(**1**) Axial and (**2**) coronal CT of patient with negative FLUS and a right-middle-lobe infiltration (arrow) on CT. (**3**) Axial and (**4**) coronal CT of patient with negative FLUS and bilateral basal infiltration (arrow) on CT. FLUS = focused lung ultrasound. CT = computed tomography.

**Figure 3 diagnostics-14-01921-f003:**
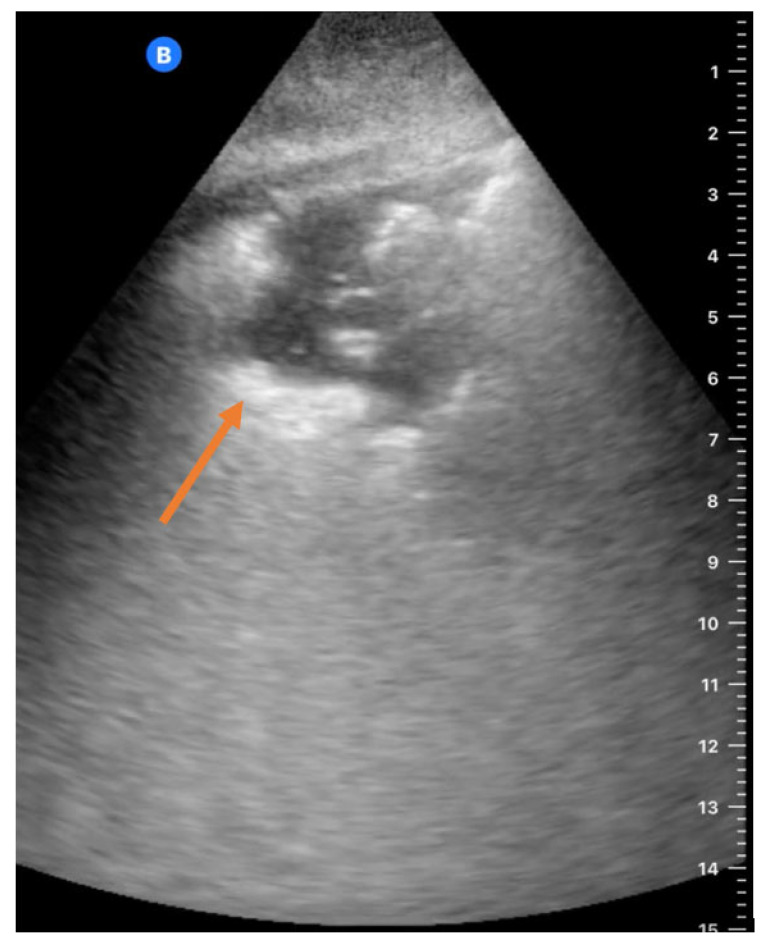
FLUS with HHUS of zone L4 with consolidation with air bronchograms (arrow). FLUS = focused lung ultrasound; HHUS = handheld ultrasound; CT = computed tomography.

**Table 1 diagnostics-14-01921-t001:** Summary of patient characteristics for the study population.

		Expert Diagnosis: Pneumonia (n = 212)	Expert Diagnosis: No Pneumonia (n = 112)	All Patients (n = 324)
**Sex (male), n (%)**		109 (51%)	71 (63%)	180 (56%)
**Age (years), mean (SD)**		73 (14)	70 (14)	71 (14)
**Current smoker, n (%)**		42 (21%)	27 (25%)	69 (22%)
**Prior smoker, n (%)**		113 (56%)	47 (44%)	160 (52%)
**Alcohol > 14 doses (12 g alcohol) per week, n (%)**		19 (9%)	15 (14%)	34 (11%)
**Nursing home resident, n (%)**		22 (10%)	6 (6%)	28 (9%)
**Comorbidities**				
**Chronic obstructive pulmonary disease, n (%)**		79 (37%)	29 (26%)	108 (33%)
**Asthma, n (%)**		32 (15%)	12 (11%)	44 (14%)
**Active lung cancer, n (%)**		7 (3%)	6 (5%)	13 (4%)
**Symptoms**				
**Cough, n (%)**		155 (76%)	66 (62%)	221 (71%)
**Sputum production, n (%)**		124 (61%)	50 (47%)	174 (56%)
**Dyspnea, n (%)**		154 (75%)	73 (69%)	227 (73%)
**Fever sensation, n (%)**		98 (46%)	51 (46%)	149 (46%)
**Fever (measured at home), n (%)**		66 (31%)	32 (29%)	98 (30%)
**Vitals**				
**Respiratory rate (breaths per minute), median (IQR)**		22 (18, 24)	20 (18, 24)	21 (18, 24)
**Heart rate, mean (SD)**		93 (19)	90 (18)	92 (19)
**Systolic blood pressure (mmHg), mean (SD)**		135 (20)	137(23)	136 (21)
**Severity score**				
**CURB-65 score, n (%)**	0	40 (19%)	33 (30%)	73 (23%)
	1	81 (38%)	29 (26%)	110 (34%)
	2	77 (36%)	38 (34%)	115 (36%)
	3	12 (6%)	10 (9%)	22 (7%)
	4	2 (1%)	2 (2%)	4 (1%)
**Blood test**				
**Leukocytes (10^9/L), median (IQR)**		12.0 (9.3, 15.2)	9.9 (7.6, 13.7)	11.3 (8.5, 14.8)
**C-reactive protein (CRP) mg/L, median (IQR)**		116 (50, 196)	54 (17, 143)	88 (37, 177)
**Mortality**				
**30 day mortality, n (%)**		8 (4%)	5 (5%)	13 (4%)

CURB-65 = Confusion, urea > 7 mmol/L, Respiratory Rate ≥ 30, Systolic BP < 90 mmHg or Diastolic BP ≤ 60 mmHg, Age ≥ 65.

**Table 2 diagnostics-14-01921-t002:** Diagnostic accuracy of FLUS and CXR for pneumonia, with different diagnostic criteria.

Expert Diagnosis					
Imaging modality	Sensitivity	Specificity	LR+	LLR−	AUC
FLUS	31% (26–36)	82% (78–86)	1.7	0.8	0.56 (0.52–0.61)
CXR	66% (61–72)	76% (71–81)	2.7	0.4	0.71 (0.66–0.76)
**CT diagnosis**					
Imaging modality	Sensitivity	Specificity	LR+	LLR−	AUC
FLUS	32% (27–37)	81% (77–85)	1.7	0.8	0.56 (0.52–0.61)
CXR	69% (63–74)	68% (62–73)	2.2	0.5	0.68 (0.63–0.74)
FLUS (excluding patients with missing posterior zones)	36% (30–42)	78% (73–84)	1.7	0.8	0.57 (0.52–0.63)
FLUS (consolidation with air bronchogram or focal b-lines was diagnostic for pneumonia)	60% (55–66)	53% (48–59)	1.3	0.7	0.57(0.51–0.62)

FLUS = focused lung ultrasound; CXR = chest X-ray; CT = computed tomography; LR+ = positive likelihood ratio; LLR− = negative likelihood ratio; AUC = area under curve. 95% confidence intervals are given in parenthesis.

**Table 3 diagnostics-14-01921-t003:** Median waiting time between admission and imaging.

	Median Waiting Time between Admission and Imaging	
Imaging Modality	Hours	Interquartile Range
Focused lung ultrasound	2.5	1.7–3.6
Computed tomography	2.6	1.8–3.7
Chest X-ray	2.0	1.5–3.0

**Table 4 diagnostics-14-01921-t004:** Median waiting time between different types of imaging.

Imaging Modality	Median Waiting Time	
	Hours	Interquartile Range
CXR and FLUS	0.6	0.3–1.5
ULD-CT/CT and FLUS	0.5	0.3–0.9
ULD-CT/CT and CXR	0.2	0.1–0.9

CXR: chest X-ray; FLUS: focused lung ultrasound; ULD-CT: ultralow-dose computed tomography; CT: computed tomography.

## Data Availability

According to Danish laws on personal data, data cannot be shared publicly. To request data, please contact the corresponding author for more information. The person responsible for the research was the principal investigator and corresponding author (MHL) in collaboration with the University Hospital of Southern Denmark. This organization owns the data and can provide access to the final data set.

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
