# Peer review of "Handheld Ultrasound Devices Used by Newly Certified Operators for Pneumonia in the Emergency Department—A Diagnostic Accuracy Study"

_diagnostics, 2024, doi:10.3390/diagnostics14171921_

Round 1

Reviewer 1 Report

Comments and Suggestions for Authors

Dear authors,

Best regards,

Comments on the Quality of English Language

Minor editing of English language is required.

Author Response

Thank you very much for your thorough and comprehensive review. Your detailed feedback is greatly appreciated and will significantly contribute to improving the quality of our manuscript.

Comments 1: Equipment: The use of the Butterfly IQ+ device may have limitations compared to high-end ultrasound machines, impacting diagnostic accuracy. Please present these in the limitation section.

Response 1:

We agree that handheld ultrasound devices, including the Butterfly IQ Plus, have limitations compared to high-end ultrasound devices. However, the main objective of this study was to evaluate the performance of handheld ultrasound devices in the hands of newly certified operators. Therefore, the limitations of these devices were not a drawback but rather the primary focus of our investigation, as discussed in detail in the main part of the discussion (lines 296 to 309). Additionally, the use of an iPhone instead of a tablet with the Butterfly IQ is mentioned under limitations, as it could potentially underestimate the accuracy of handheld ultrasound devices.

Comments 2: Equipment Evaluation: Further studies comparing different HHUS devices and their efficacy in detecting lung pathology. Please include this aspect in the future directions section.

Response 2:

Thanks for the imput we have added the following text:

”The field of HHUS is rapidly advancing with new devices continually introduced (25, 27, 42). As technology improves, their utility will likely increase. Further testing and a framework for evaluating HHUS devices are essential for progress in identifying lung pathology.”

Response to recommendation 1 and 3

Regarding recommendations 1 and 3, which suggest including a more diverse population and implementing more extensive training for FLUS operators, these are interpreted as suggestions for future studies and will be incorporated into our upcoming research. However, they have not been applied to the currently submitted manuscript

Reviewer 2 Report

Comments and Suggestions for Authors

The study assessed the diagnostic perfromance of focused lung ultrasound (FLUS) for patients with suspect pneumonia. Overall, this topic is interesting. However, I have several concerns.

1. Please add more discussion about why the sensitivity is so low. 

2. Please discuss the discrepancy between FLUS and CT. How about the involve lobe or segment.

3. English editing is needed to improve reading.

4. If possible, please suggest the algorithm incoporated both FLUS, CT and CXR for diagnosing pneumonia.

Author Response

See word file

Round 2

Reviewer 1 Report

Comments and Suggestions for Authors

Dear authors,

I agree with the modifications implemented by the authors. Upon re-verification of the text, I have observed a singular issue, namely, the presence of a 25% plagiarism rate.

A 25% match on an iThenticate report indicates that a quarter of your document's content matches sources in the iThenticate database. I provided some steps to take:

  1. Review the Matches:

    • Check the Sources: Look at which sources your text is matching. Are they credible academic sources, or less reliable websites?
    • Context of Matches: Determine if the matches are direct quotations, common phrases, or more significant sections of text.
  2. Evaluate the Nature of the Matches:

    • Quotations and Citations: If matches are direct quotes, ensure they are properly quoted and cited.
    • Common Phrases: Some phrases are common and may not be a concern, but large portions of matched text need scrutiny.
  3. Paraphrasing and Originality:

    • Improve Paraphrasing: If you've paraphrased text, ensure it's done correctly and significantly different from the original.
    • Add Original Content: Increase the proportion of your original analysis and discussion to reduce the match percentage.
  4. Check for Unintentional Plagiarism:

    • Ensure Proper Citations: Verify that all borrowed ideas and text are correctly cited.
    • Reference List: Make sure all sources are listed in your reference section.
Comments on the Quality of English Language

Minor English revision is necessary.

Author Response

Thank you for your thorough review of my manuscript. I reviewed the iThenticate report and found that Source 1, corresponding to 11%, contains scattered similarities from sources within my research departments at my universities. These similarities include the headline, generic terms, the publisher's homepage, affiliations, and statements about the institutional review board and acknowledgments. These similarities likely originate from PR materials explaining the research or from my PhD thesis, which this research is part of. Since my thesis should remain unpublished until all related articles or manuscripts are published, these similarities are not unexpected. The similarities in Source 1 mostly involve generic terms like sensitivity, specificity, pneumonia, or results, and therefore do not constitute plagiarism.

Source 2 includes similarities in the results and methods sections of the abstract of my manuscript. Since Source 2 is from the National Library in Denmark, these similarities are also likely related to my thesis. Methods and results sections are difficult to diversify and are generally not considered plagiarism.

Regarding Source 3, the similarities arise from the MDPI template used for submission, which cannot be considered plagiarism. Sources 4, 5, 6, 7, and the rest involve generic terms commonly found in studies, such as sensitivity, specificity, area under the curve, and p-values. Author affiliations, organization names, data availability statements, and authors' contributions are also not plagiarism issues.

Some similarities consist of very few words scattered throughout the text and are not significant. For example, similarity number 14 involves only seven words scattered in the methods section, which is not plagiarism.

In conclusion, we find no plagiarism issue in this manuscript, and we have not made any changes based on this report. I hope you agree with our assement.

Reviewer 2 Report

Comments and Suggestions for Authors

The authors response well.

Author Response

Thank you.

Have a nice day